# A Study on Particulate Matter from an Area with High Traffic Intensity

Dan-Marius Mustață [1], Ioana Ionel [1,*], Rareș-Mihăiță Popa [1], Ciprian Dughir [2] and Daniel Bisorca [1]

[1] Faculty of Mechanical Engineering, Universitatea Politehnica Timisoara, 300006 Timisoara, Romania; dan.mustata@student.upt.ro (D.-M.M.); rares.popa@student.upt.ro (R.-M.P.); daniel.bisorca@gmail.com (D.B.)
[2] Faculty of Electronics and Telecommunications, Universitatea Politehnica Timisoara, 300006 Timisoara, Romania; ciprian.dughir@upt.ro
[*] Correspondence: ioana.ionel@upt.ro; Tel.: +40-723349337

**Abstract:** This research focuses on analyzing concentrations of particulate matter (PM) next to a highly congested road section, with 39,900 as the maximum number of vehicles per 24 h, in the vicinity of Timisoara, Romania. The concentrations are measured in multiple episodes using two different measurement methods, gravimetric and dynamic light scattering, showing a dispersion range of the PM concentrations. The presence of metal particles in the samples are analyzed using an electron microscope. Additionally, the impact on human health is described by evaluating the results for inhalable-, thoracic-, and alveolar-sized particles.

**Keywords:** PM pollution; microscopic analysis of PM; PM influences on human health; presence of metals in PM; inhalable-, thoracic-, and alveolar-sized particles





## 1. Introduction

The primary objective of this article is to present the findings pertaining to the levels of particulate matter (PM) in close proximity to a heavily trafficked road segment. This investigation entailed a comparative analysis of two distinct measurement methodologies, alongside an examination of the metallic chemical particles identified within the collected samples.

Maintaining clean air is a crucial matter for every government worldwide, particularly for developed nations that prioritize implementing comprehensive approaches to mitigating air pollution, especially in densely populated urban regions. The negative impact of pollution and inferior air quality in cities can lead to a host of health issues, predominantly cardiovascular and respiratory ailments. In accordance with a 2014 World Health Organization (WHO) report, a staggering 80% of premature deaths caused by heart attacks and severe heart diseases are linked to air pollution levels [1].

The detrimental effects of elevated air pollution on human health were first recognized at the start of the 20th century following severe environmental pollution incidents. Notable examples include the Meuse Valley in Belgium (1930), London in Great Britain (1952), and Donora in the United States (1948). These events led to the observation of adverse health effects on individuals, ranging from headaches and vomiting to fatalities [2].

Internal combustion engines produce a variety of pollutants during the combustion process, including suspended particles of varying sizes ($PM_1$, $PM_{2.5}$, and $PM_{10}$). These particles can result in bronchitis, asthma, and inflammation of the respiratory tract [3]. Gases also contribute to health issues like carbon oxides ($CO$ and $CO_2$) by diminishing the blood's capacity to transport oxygen [4] and nitrogen oxides ($NO$ and $NO_2$) which are responsible for respiratory diseases, particularly asthma [5].

Particle matter pollution is a long-known hazard to human health, and it continues to be to this day an important trigger of serious pulmonary and cardiovascular diseases [6],

contributing to a great number of premature deaths per year, making it a leading cause of mortality worldwide [7].

This section will focus more on the respiratory effects of these particles.

An important factor is the size of the particles, as this is directly linked to how much damage it causes to the tissues it manages to reach.

It is widely known that the smaller the particle, the deeper it can penetrate the respiratory tract causing important inflammatory effects along with oxidative stress on the tissues it reaches [8]. The body's natural barriers can limit, to a certain degree, the intake of these particles, as the hair and mucus from the upper respiratory tract can trap and further eliminate larger particles from the body, whereas smaller particles can reach as far as the alveoli and even into the bloodstream [9].

When fine particles penetrate deeply into the lungs, they trigger reactions, including oxidative stress caused by excessive free radicals from the particles overpowering the body's antioxidant defenses. This accentuates inflammatory responses and harms the affected tissues [10]. $PM_{10}$ particles can be filtered by the nasal mucous layer and cilia, but longer exposure can lead to local inflammatory effects by irritating the nasal cavity which, in turn, can damage the protective barrier of the nose [11]. Several conditions can arise or become aggravated because of this, one example being allergic rhinitis [12]. The trachea possesses its own protective barrier, producing mucus, and lined with cilia that move to expel trapped particles, preventing them from advancing into the respiratory tract. The trachealis muscle also aids in particle expulsion through its contractions [13]. Chronic exposure, or continuous and repeated exposure to coarse particles, leads to local inflammation of the trachea, thus disrupting its protective barrier causing a number of harmful effects and conditions like acute and chronic laryngitis [14]. Although it is still unclear if long-term exposure (months and years) to $PM_{10}$ particles can lead to laryngeal cancer, several articles have concluded that this risk is greater when it comes to fine particles ($PM_{2.5}$) rather that coarse particle ($PM_{10}$) [15].

Other organs that are directly affected by $PM_{10}$ particles are the eyes, where exposure initiates several inflammatory responses, causing dry-eye-like symptoms [16]; long-term exposure can lead to a chronic condition where the eyes do not produce enough lubrication, thus increasing the risk of infection and damage to the surface of the eyes, which in extreme cases can even lead to vision loss [17].

Fine particulate matter ($PM_{2.5}$) bypasses initial protective barriers and enters the lower respiratory tract. This triggers oxidative stress upon contact with lung tissues, generating an excess of free radicals that overwhelms the body's antioxidants. Consequently, lung cells, DNA (deoxyribonucleic acid), and DNA repair are damaged, elevating the risk of a severe outcome: lung cancer [18]. The lung cells suffer damage through an inflammatory response triggered by particle contact. Maintaining a balance between inflammatory and anti-inflammatory factors is crucial for normal lung function. Chronic exposure to these particles leads to persistent and excessive inflammation, disrupting lung homeostasis and causing uncontrolled tissue damage [19]. Uncontrolled chronic lung inflammation induces the production of growth factors, enzymes, and ECM factors, leading to irreversible scarring tissue, pulmonary fibrosis, and an elevated risk of lung cancer [20].

As several studies show, acute inflammation of the lungs due to the inhalation of $PM_{2.5}$ particles can exacerbate symptoms of preexisting conditions such as asthma, increasing the number of hospital admissions and deaths due to acute asthmatic episodes [21].

Although $PM_{2.5}$ can also be absorbed into the blood stream, ultrafine $PM_1$ particles are the most harmful, as they have a greater chance of breaking the blood barrier because of their smaller size. After reaching the alveoli, a majority of them pass the cell membrane, entering the blood stream and spreading to other organs of the body, causing further damage [22]. Once fine particulate matter breaches the blood–brain barrier, it becomes challenging to eliminate, resulting in neuroinflammation and cerebral dysfunction. Recent studies have linked particulate matter pollution to neurological conditions such as cognitive decline and dementia [23].

Ultrafine particles entering the bloodstream trigger vascular inflammation, releasing excessive inflammatory factors affecting blood coagulation and vessel integrity. Prolonged exposure to these particles raises the risk of developing atherosclerosis, a significant contributor to global cardiovascular mortality [24]. Several studies showed that long-term (months and years) exposure is also linked to an increased risk for irregular heart rate, hypertension, heart failure, and even stroke [25].

All of the above represent serious effects on human health and well-being. That is why it is of extreme importance to find ways to reduce air pollution, especially particulate matter pollution, because it represents one of the most harmful existing elements that greatly impacts human health worldwide.

Typically, traditional air quality measuring devices are utilized by governments and state institutions, which are placed at various locations throughout urban and peri-urban regions. These devices are located in stationary buildings and necessitate substantial energy consumption and frequent maintenance to function. However, technological advancements in air quality measurement have introduced novel techniques that make the measuring equipment portable, thereby enabling easy mobility and mobile usage [26].

Installing an air quality monitoring system in public transport stations provides a targeted approach for analyzing and addressing high pollution levels that directly affect passengers waiting at these stops. Although state institutions monitor air quality in urban areas with high population densities, they do not specifically address the unique situations encountered by public transport passengers. While personal vehicle passengers are shielded from pollution by advanced automotive technology, passengers waiting at public transport stops are constantly exposed to harmful pollutants generated by nearby traffic.

The presence of filters in vehicles reduce the introduction of suspended particles, such as $PM_{2.5}$ and $PM_{10}$, into passenger compartments of vehicles [27], providing a safe environment for drivers, but those waiting in public transport stations are constantly bombarded by passing vehicles. This exposure disproportionately affects vulnerable groups, such as the elderly and children, who rely heavily on public transport for their daily commutes. Furthermore, a plethora of scientific investigations have revealed that despite the presence of protective filters that effectively inhibit the ingress of particulate matter into the passenger compartment, the act of operating a vehicle with the windows open leads to a significant amplification of PM concentrations, with an increase of up to 80% being observed in various studies [28].

To measure the levels of suspended particles and polluting gases emitted by motor vehicle thermal engines, air quality measurement equipment was utilized. Advanced people counting systems, which rely on video recording systems placed in specific areas, were employed to determine the number of passengers exposed to the pollutants [29]. The main objective of this study was to assess the concentration levels of suspended particles, which are known to have a detrimental impact on human health. The proposed measurement system continuously monitored the air quality and passenger counts at public transport stops. This research builds on a previous study conducted on the same road section but with a shorter duration of PM value recordings.

The present study covered a period of more than 20 days and is discussed in detail in this article [30].

## 2. Materials and Methods

The concentration of particulate matter was determined using two different measurement methods with two separate pieces of equipment. Both instruments were calibrated, metrologically certified, and accredited by the RENAR (Accreditation Association of Romania).

Two different sets of results were recorded. Over a 23-day episode between 15 July 2022 and 7 August 2022, measurements were taken using gravimetric impactors. Additionally, on 25 September 2022, measurements were taken using both the DLS method and gravimetric impactors in parallel.

The first measurement equipment consisted of gravimetric impactors that measure particulate matter concentrations through weight comparisons of charged versus uncharged fiber glass filters. These impactors, produced by Sven Leckel GMBH, Berlin, Germany, under the model's name LVS3 (Low Volume Sampler), are specially designed to aspirate air through constructed nozzles at a rate of 55 $Nm^3$/24 h. During a 24 h measurement range, particles of a specific dimension are captured on a glass fiber filter. The charged filter is then weighed and compared to the initial state of the uncharged dry filter, determining the concentration of particulate matter of the targeted dimension. The measurement equipment is in accordance with the European standard CEN EN 12341 [31].

The determination of the concentrations is performed by weighing the charged filters (Ø 40 mm) and subtracting the initial value of the uncharged filter. The result in grams is divided by the air flow measured in 24 h that is aspired through the impactor nozzle and multiplied by $10^6$ to reach a result measured in $\mu g/m^3$. The measurements were taken under normal temperature and pressure conditions.

This measurement method allows for microscopic analysis of the captured particles because of their deposition on the filters, which is an analysis that is presented in this research paper. The microscopic analysis was performed with the Quanta FEG-250 SEM produced by ELECMI, Zaragoza, Spain, which is an environmental scanning electron microscope used for high-resolution imaging, and the composition analysis used energy-dispersive X-ray microanalysis (EDS). A section of the fiber glass filter was used under microscope to visualize the dust particles at a 10,000 times magnification which shows different sized particles below 10 $\mu m$.

The second measurement equipment was a spectrometer using light scattering to determine the concentration of particulate matter, allowing for measurements to be taken every second, leading to detailed concentration readings over the timespan of the 24 h measurement period. This equipment is produced by GRIMM Aerosol Technik, part of Durag Group, Hamburg, Germany.

Measuring the fluctuations in intensity of the light scattered from a suspension or solution allows for the determination of the particle size. This method is widely known as dynamic light scattering (DLS), although it is also referred to as photon correlation spectroscopy (PCS) and quasi-elastic light scattering (QELS). The latter terms are more frequently encountered in older literature. DLS finds its primary application in the analysis of nanoparticles. It is commonly employed for tasks like determining the size of nanogold, proteins, latex particles, and colloids. Typically, this technique is most effective for submicron particles and can accurately measure particle sizes below a nanometer. Within the range of microns to nanometers, when it comes to size measurement (excluding thermodynamics considerations), the distinction among molecules (e.g., proteins or macromolecules), particles (e.g., nanogold), and even a secondary liquid phase (e.g., emulsion) becomes less clear. While DLS can be utilized as a tool to investigate complex fluids like concentrated solutions, this particular application is relatively less common compared to particle sizing [32].

The measurement sampling was undertaken within a locale characterized by a significant volume of vehicular activity, with the maximum recorded count of vehicles amounting to 39,900 vehicles per 24 h [33]. This investigation was carried out in close proximity to the city of Timisoara, Romania, specifically along the DJ691 (road indicator), serving as a crucial link connecting the A1 highway to Timisoara. The designated measurement site was situated adjacent to a public transportation stop within a residential region, where the concentrations of particulate matter (PM) exert a substantial influence on the well-being of the local residents. This road is subject to high traffic volumes due to several factors, including its location on the outskirts of Timisoara, a major Romanian city; its role as the primary access road to highway A1; and its proximity to production facilities and logistics hubs. These conditions result in a significant number of vehicles traveling along this main road section. Given the small surface area of Dumbravita and its status as a residential area,

most public transport stops are located alongside this one-lane road with 11 such stops in total.

## 3. Results

The results are presented for different measurement methods and episodes, including a microscopic analysis of the particles caught on the filters used in the gravimetric method measurement.

### 3.1. Determination of Concentration of $PM_{2.5}$ and $PM_{10}$ Using Gravimetric Impactors

The concentrations of $PM_{2.5}$ and $PM_{10}$ were measured over a 23-day period, with each day representing a 24 h measurement cycle.

The concentrations of $PM_{2.5}$ and $PM_{10}$ were directly proportionate to the recorded temperatures, as shown in Figure 1.

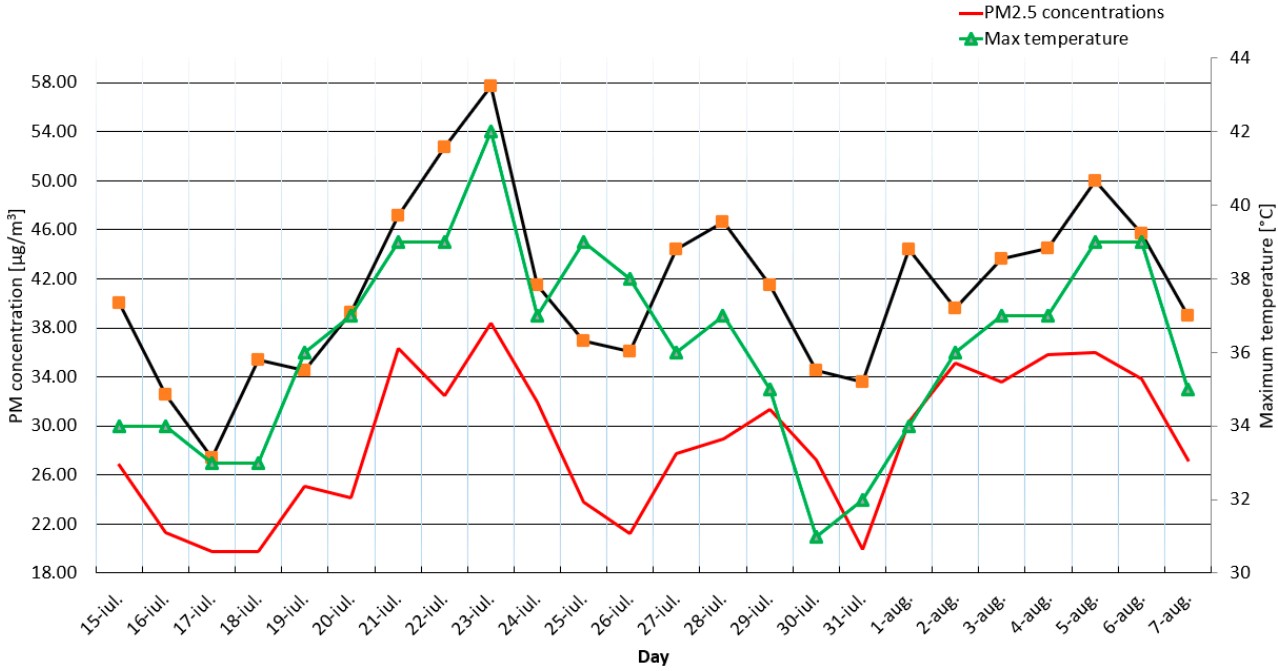

**Figure 1.** Daily concentrations of $PM_{2.5}$, $PM_{10}$, and air temperature recorded over the 23-day measurement period.

The highest concentration for both $PM_{2.5}$ (38.43 µg/m$^3$) and $PM_{10}$ (57.71 µg/m$^3$), the highest number of vehicles (39,900), and the highest temperature (42 °C) were recorded on 23 July 2022.

The standardized number of vehicles was determined by measuring, each 15 min, a one-minute interval count of the vehicles over the entire 24 h episode. Each vehicle was standardized based on the size and mass of the different vehicle types (cars, lorries, trucks, and busses). Figure 2 shows the measured number of vehicles.

It can be recognized that the trend in the number of vehicles was directly proportionate to the concentration levels of PM.

Wind speeds were low, at an average of 10 km/h over the 24 h period, with the wind direction blowing from N-E, perpendicular to the road section, directly towards the measurement equipment. All these factors lead to the recording of high concentrations of $PM_{2.5}$ and $PM_{10}$ during that day [34].

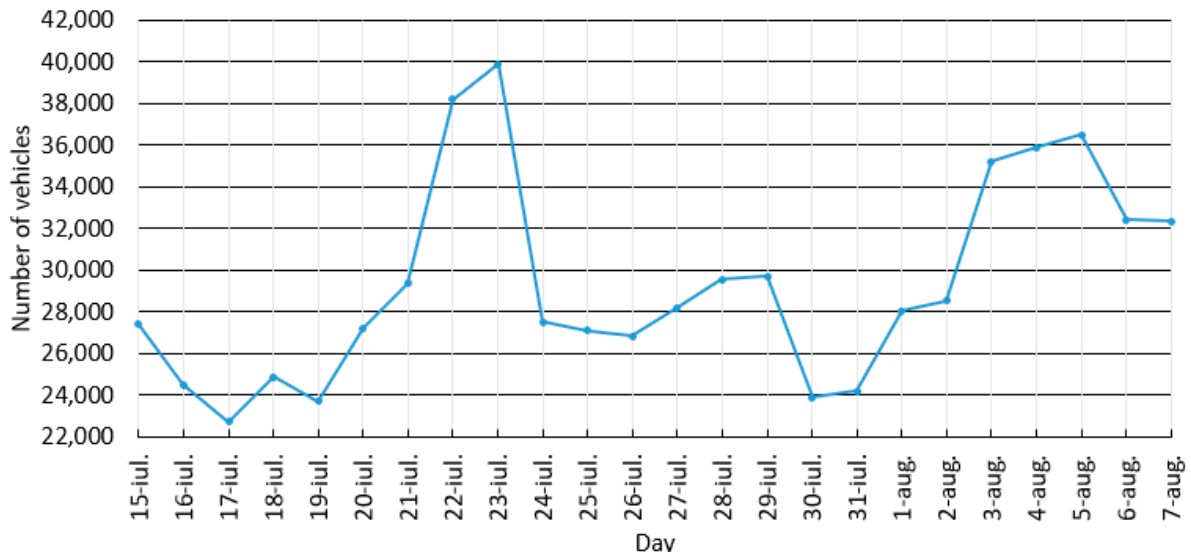

**Figure 2.** Standardized number of vehicles recorded over the 23-day measurement period.

During heat waves and dry weather conditions, particulate pollution increases. The soil becomes very dry, which also results in an increase in dust particles being raised into the atmosphere [34].

The highest atmospheric pressure recorded was at 1007 mbar, which is in correlation with the heat wave temperature recorded at 42 °C.

Through an analysis of the air temperature, atmospheric pressure, and wind speeds, it is possible to account for the elevated levels of PM. During heat waves and periods of low atmospheric pressure and weak wind activity, air movement is restricted, which allows particulate matter to accumulate and remain within the region rather than being dispersed [35].

### 3.2. Determination of Concentrations of $PM_{2.5}$ and $PM_{10}$ Using DLS Spectrometer

On 25 September 2022, the $PM_{2.5}$ and $PM_{10}$ concentrations were measured using DLS technology with a spectrometer. This cutting-edge technology enables measurements to be taken every second throughout an entire 24 h period, providing highly accurate data that can reveal variations in the readings at different times of the day. The concentrations of $PM_1$, $PM_{2.5}$, and $PM_{10}$ measured using the spectrometer are presented in Table 1.

**Table 1.** Values of $PM_1$, $PM_{2.5}$, and $PM_{10}$ concentrations measured using the DLS spectrometer.

| Variable | $PM_1$ (µg/m³) | $PM_{2.5}$ (µg/m³) | $PM_{10}$ (µg/m³) |
|---|---|---|---|
| Minimum | 9.9 | 10.9 | 13.5 |
| Maximum | 130.5 | 134.6 | 149.9 |
| Mean | 20.9 | 22.7 | 29.3 |

Temperatures were recorded on 22 September 2022 at an average of 14 °C, with a maximum of 21 °C and minimum of 7 °C. The wind speeds were on average at 8 km/h [36].

Using the impactors, the concentrations were shown to be 30.03 µg/m³ for $PM_{2.5}$ and 32.27 µg/m³ for $PM_{10}$ for the 24 h measurement period, which are close to the concentrations measured with the spectrometer.

The recorded concentrations of PM over the 24 h measurement period are shown in Figure 3. The mean values of these concentrations for $PM_1$, $PM_{2.5}$, and $PM_{10}$ are indicated in Figure 4.

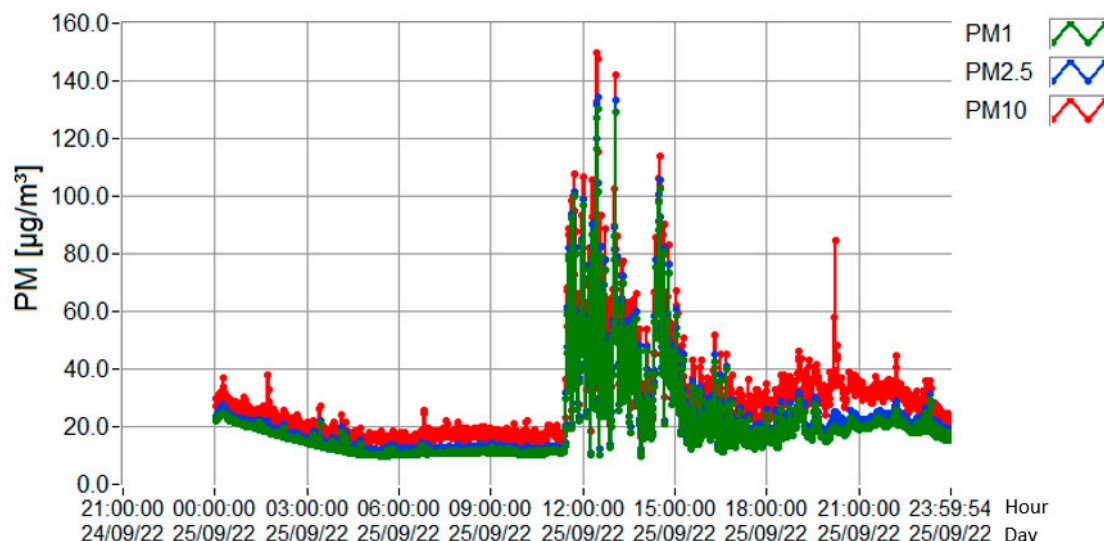

**Figure 3.** Concentrations of PM$_1$, PM$_{2.5}$, and PM$_{10}$ recorded over the 24 h measurement period with samplings at each second.

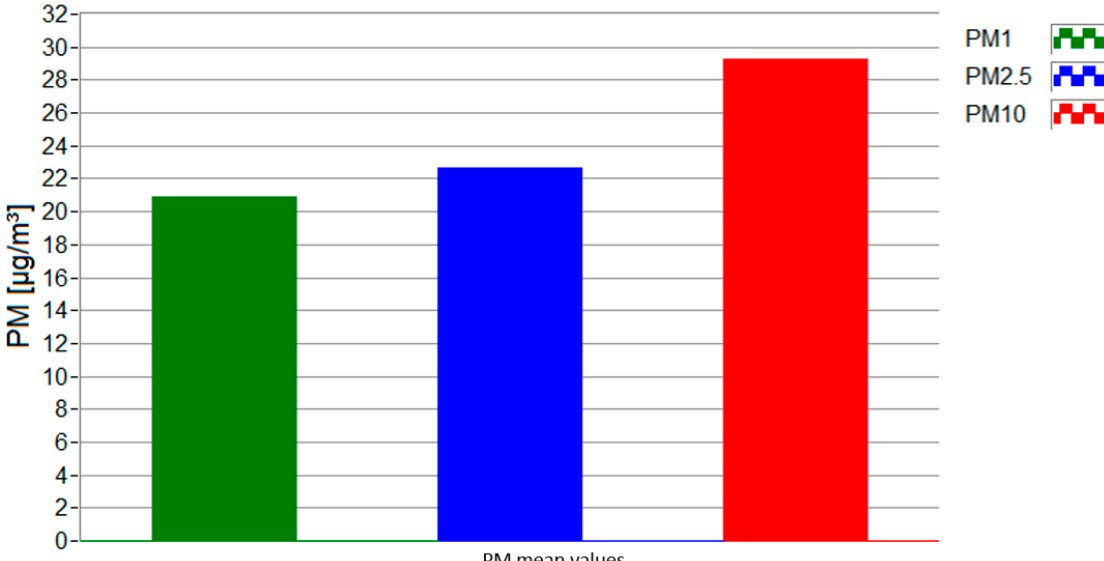

**Figure 4.** Mean values for PM$_1$, PM$_{2.5}$, and PM$_{10}$ concentrations recorded over the 24 h measurement period.

### 3.3. Determination of Concentrations of Inhalable-, Thoracic-, and Alveolar-Sized Particles

Using the spectrometer, it was possible to determine the concentrations of inhalable-, thoracic-, and alveolar-sized particles by analyzing the levels of PM$_1$, PM$_{2.5}$, and PM$_{10}$. Since different particle sizes can penetrate various parts of the human body, this advanced technology can provide a more comprehensive understanding of the potential health impacts associated with each size category. All particle sizes are inhalable, but only PM$_{2.5}$- and PM$_1$-sized particles reach the thoracic level and, finally, only PM$_1$ is able to reach the alveoli. In Figure 5, the different concentrations recorded over the 24 h measurement period are shown. The mean values for the concentration of air quality (AQ) are shown in Figure 6.

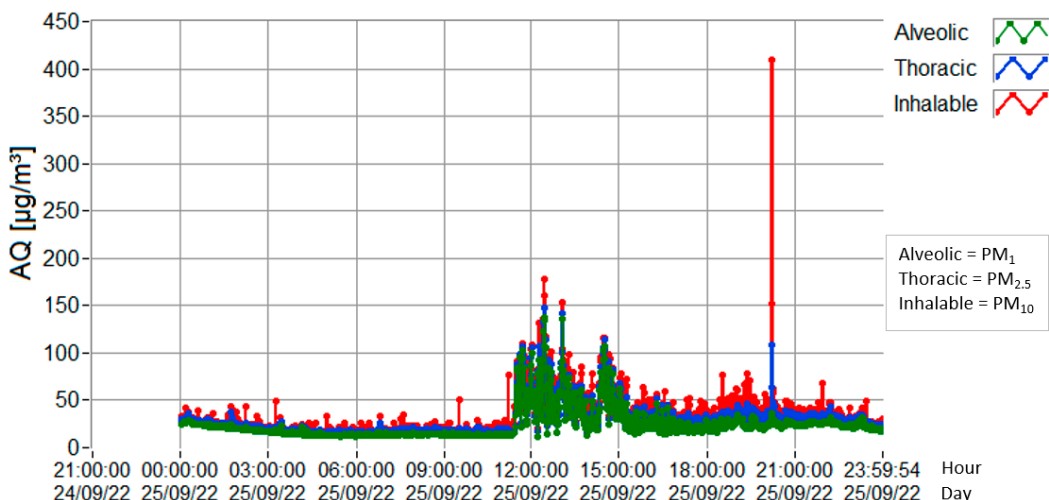

**Figure 5.** PM concentrations transposed to inhalable-, thoracic-, and alveolar-sized particles.

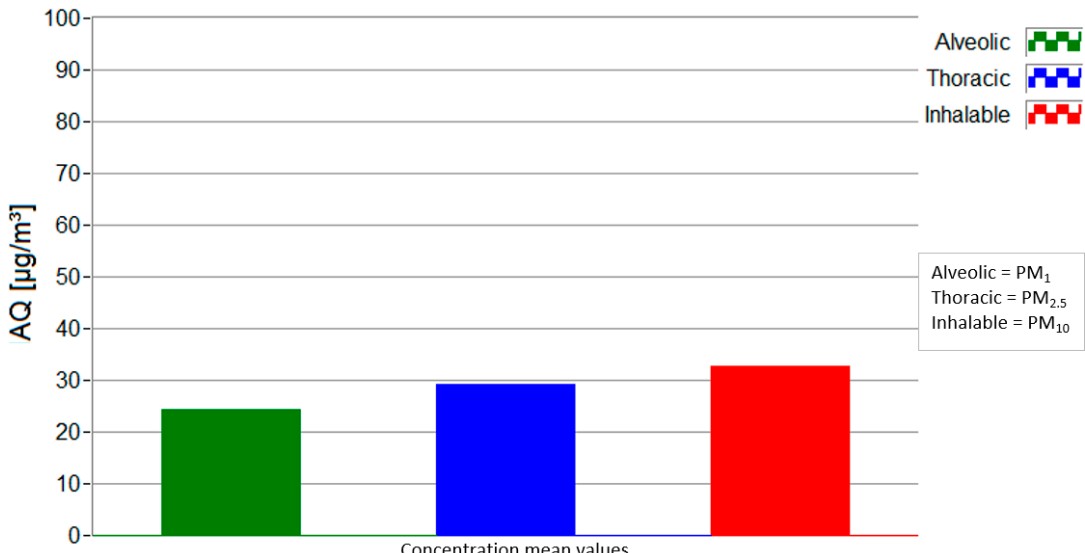

**Figure 6.** PM concentrations' mean values transposed to inhalable-, thoracic-, and alveolar-sized particles.

*3.4. Microscopic Analysis of Particles Captured with Gravimetric Impactors*

Gravimetric impactors utilize glass fiber glass filters to capture particles of varying sizes, as they are drawn into the apparatus from the surrounding air. Following deposition on these filters, the particles are subjected to electron microscope analysis. Analysis of the samples revealed the presence of various chemical elements, including O, C, Na, Mg, Al, Si, Mo, Ca, and Fe, each present in different proportions. The ratio of each element found on the filter of $PM_{2.5}$ is shown in Figure 7.

In comparison, the $PM_{10}$ filter deposited all elements present on the $PM_{2.5}$ filter, except for Mo, and with the inclusion of an iron (Fe) percentage.

Carbon (C) was the most prevalent chemical compound in both the $PM_{2.5}$ and $PM_{10}$ samples, together with oxygen ($O_2$), as a result of burning fuel in the internal combustion process. In terms of concentrations, carbon showed a smaller percentage in $PM_{2.5}$ than in the $PM_{10}$ samples because of the size of the particles, and $PM_{10}$ was able to catch larger particle sizes. In regard to metals, it can be observed that iron (Fe) was present only in the $PM_{10}$ samples compared to molybdenum (Mo), which had a presence only in the $PM_{2.5}$ samples.

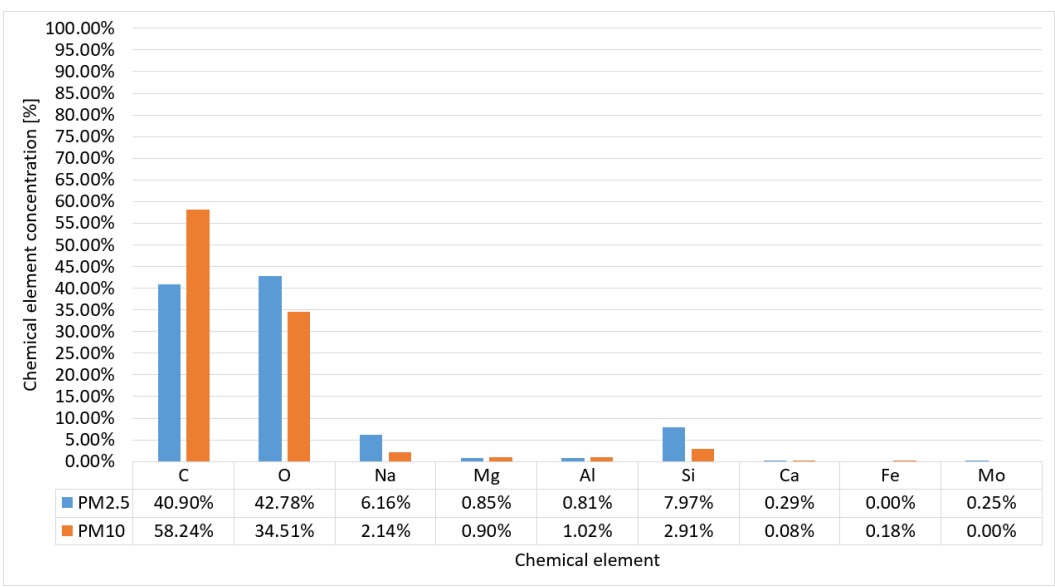

**Figure 7.** Percentage of each chemical element present in the analyzed $PM_{2.5}$ filter.

Taking a closer look at the metal particles, comparing between the different-sized PM samples ($PM_{2.5}$ versus $PM_{10}$), sodium (Na) showed a greater difference in concentrations between $PM_{2.5}$ and $PM_{10}$, with a greater presence in the $PM_{2.5}$ samples. The rest of the metal particles, like magnesium (Mg), aluminum (Al), Fe, and calcium (Ca), showed smaller differences in the concentrations among the PM sample sizes.

An image of the microscopic analysis is presented in Figure 8, where the size and shape of the different particles and how they are deposited onto the filter fibers can be observed.

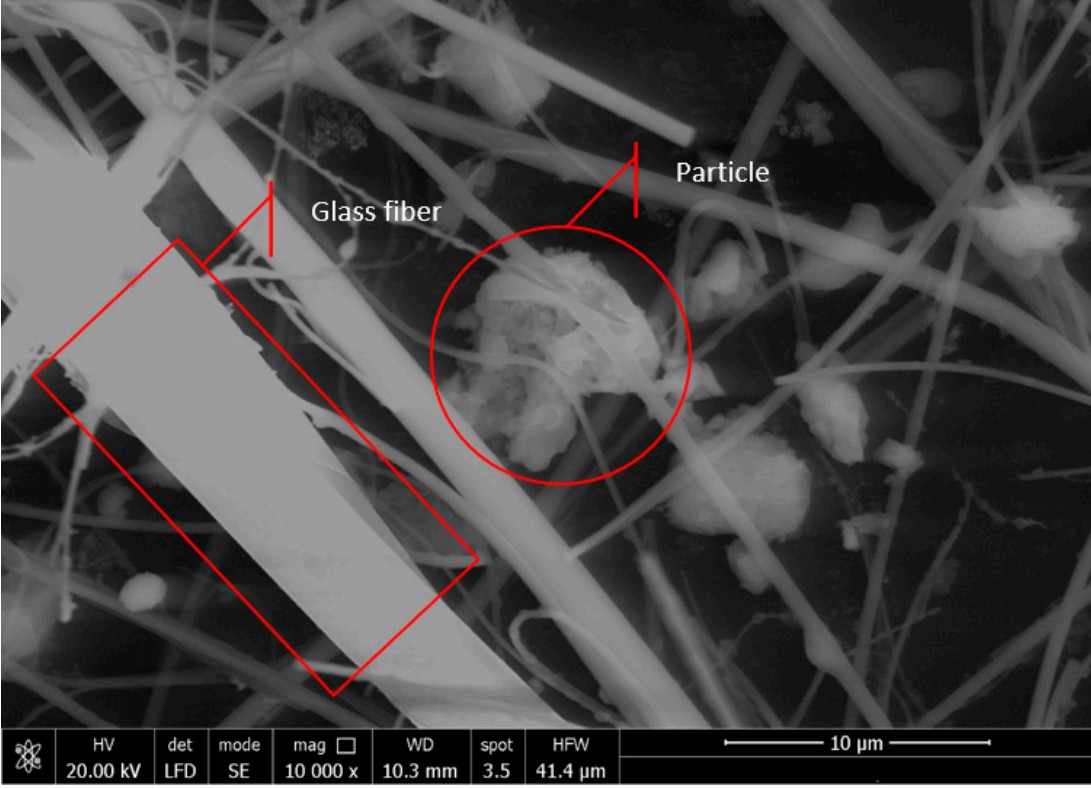

**Figure 8.** Magnified view of dust particles caught between the glass fibers of the filters (extract from $PM_{10}$ filter).

## 4. Discussion

The purpose of this study was to show the concentrations of PM near a highly trafficked road section. The novelty of this research lies in the utilization of two distinct measurement methodologies to ascertain the levels of particle concentrations. The authors, drawing upon their experiential insights, deem the incorporation of complementary measurement devices and methods as highly significant in this context.

Under Romanian Law, the annual limit for $PM_{10}$ concentration is set at 40 $\mu g/m^3$, with a daily limit of 50 $\mu g/m^3$ that cannot be exceeded for more than 35 days within a given year. However, for $PM_{2.5}$, Romanian Law does not cover any limits compared to the European Union Directive 2008/50/EC, which states an annual average limit for $PM_{2.5}$ of 20 $\mu g/m^3$ [37,38]. For $PM_1$, neither Romanian Law nor European Union Directives state any limits.

Comparing the data obtained over the 23-day period using gravimetric impactors to that obtained over the 1-day parallel measurement using both gravimetric impactors and a spectrometer, it is evident that the $PM_{10}$ concentrations exceeded the daily limit of 50 $\mu g/m^3$ on 13 out of the 23 days with the highest recorded value of 57.71 $\mu g/m^3$ recorded on 23 July 2022. While these fall short of the annual limit of 35 days, it is worth noting that exceeding the limit on 13 out of 23 days suggests that the limit is likely to be breached in the longer term, thereby contravening relevant legislation. Although current legislation does not address $PM_{2.5}$ and $PM_1$ concentrations, these particles are known to pose a greater threat to human health than $PM_{10}$, as explained in the article. Therefore, it is essential to consider these finer particulate matter sizes when assessing air quality and associated health risks, even if they are not explicitly regulated by existing laws.

With regard to the chemical analysis conducted on PM filters, it is evident that carbon emerges as the prevailing chemical element in terms of abundance. This outcome is to be anticipated, considering carbon's role as a byproduct of combustion processes inherent in internal combustion engines. Hence, the observed proportion of carbon is in line with expectations.

## 5. Conclusions

This research concludes the presence of particulate matter in dangerous concentrations with values for PM between 18 and 58 $\mu g/m^3$ based on the gravimetric measurement method, as well as values between 20 and 40 $\mu g/m^3$ based on the DLS measurement method. According to the medical literature, particulate matter in high concentrations poses a threat to human life and the environment. Particulate matter samples collected using gravimetric impactors possess a distinct advantage in their applicability to microscopic analysis for identifying chemical compounds, such as metals. The measurement methodology employed by gravimetric impactors involves the deposition of particles onto filters, facilitating their subsequent examination under a microscope to determine their chemical composition. On the other hand, spectrometers do not deposit the particulate matter onto surfaces and, as a consequence, lack the capability to directly detect the chemical composition of the captured particles. However, spectrometers offer notable benefits, including the ability to measure concentrations of inhalable, thoracic, and alveolar particles by leveraging their respective size ranges. This research study, characterized by its inherent complexity, yielded valuable insights not only into the determination of concentrations of particulate matter (PM) near a high traffic intensity area but also into the realm of human health effects associated with particulate pollution. By employing multiple measurement methods and meticulously analyzing the obtained data, this study has provided a comprehensive understanding of the intricate relationship between PM concentrations and their potential impacts on human well-being. By addressing both the determination of PM concentrations and their impact on human health, this research study provides a comprehensive framework for understanding the multifaceted nature of particulate pollution. These insights contribute to the broader field of environmental health, assisting policymakers and public

health professionals in formulating effective strategies to mitigate the adverse effects of particulate pollution in areas characterized by high traffic intensity.

**Author Contributions:** Conceptualization, D.-M.M.; methodology, D.B.; software, C.D.; validation, I.I.; formal analysis, I.I.; investigation, R.-M.P.; resources, I.I.; writing—original draft preparation, D.-M.M.; writing—review and editing, I.I. All authors have read and agreed to the published version of the manuscript.

**Funding:** This research received no external funding.

**Institutional Review Board Statement:** Not applicable.

**Data Availability Statement:** Data is contained within the article as variable values and graphical representations.

**Acknowledgments:** All authors wish to convey heartfelt appreciation to Diana Raluca Streinu, for her full availability, and for offering explanations on the medical terms and information contained within this article.

**Conflicts of Interest:** The authors declare no conflict of interest. The funders had no role in the design of the study; in the collection, analyses, or interpretation of data; in the writing of the manuscript; or in the decision to publish the results.

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
