# Peer review of "A Study on Particulate Matter from an Area with High Traffic Intensity"

_applsci, doi:10.3390/app13158824_

Round 1

Reviewer 1 Report

This article presents results on particulate matter (PM) levels in the vicinity of a high-traffic road segment, in the vicinity of Timisoara, Romania. Concentrations have been measured using two different measurement methods: gravimetric and dynamic light scattering. Using an electron microscope was analyzed the presence of metal particles in the samples. In addition, the impact on human health is described by evaluating the results for inhalable, thoracic and alveolar particles. 

The article is very interesting and analyses the effects of PM on human health, the organs affected by the particles and the mechanisms of lung cell damage. In general, there are no major problems, but some recommendations are provided.

The following typing errors are reported: 

·       PM1 is best indicated in the same way, PM1 instead of PM1.0 (change all occurrences); 

·       PM1, PM2.5 and PM10 should be indicated with a subscript and written as “PM₁, PM₂.₅, PM₁₀”;

·       Lines 38-39: Chemical formulae must be written with subscript. CO2 has to be corrected with CO2 and NO2 with NO2;

·       Line 58: remove the space between PM and 10;

·       Line 97: remove the space between PM and 2.5 and write PM in capital letters;

·       Line 222 (Figure 1): For this graph, one could insert another axis for temperature, so as not to have different units on the same axis.

·       Line 287: “In comparison, PM10 filter deposited all elements present on the PM2.5 filter, with an inclusion of Fe percentage as shown in Figure 7.” The sentence turns out to be true although not entirely accurate, as Mo is not present in the PM10 filter; as shown in line 295. 

·       Line 318: the statement is not quite correct because in April 2008 the European Union definitively adopted a new directive (2008/50/EC) that sets air quality limits with reference also to PM2.5, which until then had not been taken into account, despite being the most dangerous to our health. However, the fact remains that no daily limits are set for PM2.5 as for PM10, but only annual limits. This limit value was set at 25 µg/mc as an annual average.

The paper provided a comprehensive understanding of the intricate relationship between PM concentrations and their potential impact on human well-being. 

As the use of complementary methods to measure PM concentrations is extremely important and significant, it is right to recognise that the novelty of this research lies in the use of two distinct measurement methodologies to ascertain particle concentration levels.

Author Response

Dear Reviewer 1,

Thank you for taking the time to review our manuscript and for your suggestions.

Your insights have undoubtedly enriched the quality of our work.

Please know that we hold your expertise and judgment in high regard.

Please see the attachment for a point-by-point response to your comments.

With respect and appreciation,

ing. Dan-Marius Mustata

Reviewer 2 Report

1. There is no adequate description of the collection of particulate matter.

2. Size of filters.

3. It does not present data on the meteorological conditions to verify the conditions it presents.

4. Does not present a description of the gravimetric process.

5. What are the disadvantages of using DLS? It only presents results from one day of sampling with this methodology.

6. Because it was sampled on those days.

7.- An adequate description of the detection equipment is not presented.

8.- The units used internationally are µg/m3.

9.- The titles of the tables and figures do not have an adequate description.

10. In general, improve the order of the experimental part.

11. The discussion of the results is confusing.

12.- It does not present the analysis of vehicular traffic.

13. Does not present the description of the electron microscope.

14.- It is necessary to relate the title, with the objective and the results

Author Response

Dear Reviewer 2,

I wanted to extend my heartfelt gratitude to you for dedicating your time to reviewing our manuscript and providing us with your valuable suggestions. Your insights have undeniably enriched the quality of our work, and we genuinely appreciate the effort you put into the review process.

Your expertise and judgment are held in high regard by all of us involved in this project. Your constructive feedback has been invaluable, guiding us towards refining and enhancing the content to ensure its scholarly excellence.

Please see the attachment with a point-by-point response to your comments.

With respect and appreciation,

ing. Dan-Marius Mustata.

Reviewer 3 Report

The article presents some data collected for different PM fractions. The article needs some english corrections. Some specific comments:

- Why not using the common used units for air particle concentrations?  (e.i. µg m-3 )

- In the introduction: from line 45 until line 118, it needs to be reduced, this article is not about the effect of particles in human health, there is no data comparing the results with some medical reports for example. This section can be reduced to a couple of sentences.

- Results section, 3.1, lines 214-219: this is more like method section.

- Line 252: instead of high/low can you use maximum and minimum temperatures?

- All graphs needs improvement: X axes are difficult to read, they look like a screenshot. 

- Figure 4: it's better to add to the legend which PM you are representing (PM2.5, PM10, etc.), having it related with the part of the body that it might affect it's quite difficult to remember, you can specify that in the legend though.

- Figure 6,7 and 8 can be combined in a single table and it will be easier to compare when describing the results.

- Figure 9: can you add more info to the image? indicating what's a fiber and what's a particle. This will be great for those like me that doesn't work with these images.

- Conclusions: you haven't used your data and compare with medical records, so you can't conclude that the data obtained are good to study the impacts on human well-being (lines 355-363

The article needs some english correction, sometimes the grammar is correct but the technical terms are wrong.

Author Response

Dear Reviewer 3,

Thank you for reviewing our manuscript and providing valuable suggestions. Your insights have enhanced the quality of our work. We greatly respect and appreciate your expertise and judgment.

In relation to your suggestion to remove the medical information, typically, articles addressing pollution tend to concentrate on either pollution measurement results, the impact on human health, or the environmental consequences. Integrating both the measured PM concentrations and health effects, along with an in-depth examination of the mechanisms by which PM influences human health, provides a comprehensive visualization and direct understanding of the cause-and-effect relationships. This holistic approach is of paramount significance in relation to human well-being. It is also in alignment with the title and expectation of this Special Issue Journal.

Nevertheless, as recommended, we have reduce the content where senseful.

Please see the attachment with a point-by-point response to your comments.

With respect,

ing. Dan-Marius Mustata.

Reviewer 4 Report

Dear Authors,

changes and suggestions would be found in the attached PDF-file, in the specific comments for all sections of the paper.

Best regards

Reviewer

Author Response

Dear Reviewer 2,

I am writing to express my heartfelt appreciation for dedicating your valuable time to review our manuscript. Your suggestions and insights have been incredibly valuable and have undoubtedly enhanced the quality of our work.

Your expertise and judgment are held in high regard by us, and we are grateful for the constructive feedback you provided. Your thoughtful comments have guided us in refining the manuscript and have contributed significantly to its overall improvement.

Once again, thank you for your dedication to the peer review process and for being an essential part of advancing our research. Your contributions have been instrumental, and we are honored to have had you as a reviewer for our work.

Please see the attachment with replies to each of your comment.

With utmost respect and appreciation,

ing. Dan-Marius Mustata

Round 2

Reviewer 2 Report

The article presents a great improvement and greater understanding

Reviewer 3 Report

Thanks to the authors for improving the article. I think it reads much better now.